# Palmitoleic Acid Acts on Adipose-Derived Stromal Cells and Promotes Anti-Hypertrophic and Anti-Inflammatory Effects in Obese Mice

**DOI:** 10.3390/ph15101194

**Published:** 2022-09-28

**Authors:** Jussara J. Simão, Maysa M. Cruz, Fernanda M. Abdala, Andressa Bolsoni-Lopes, Lucia Armelin-Correa, Maria Isabel C. Alonso-Vale

**Affiliations:** 1Post-Graduate Program in Chemical Biology, Institute of Environmental Sciences, Chemical and Pharmaceutical, Federal University of Sao Paulo, Diadema 09913-030, SP, Brazil; 2Department of Biological Sciences, Institute of Environmental Sciences, Chemical and Pharmaceutical, Federal University of Sao Paulo, Diadema 09913-030, SP, Brazil; 3Department of Nursing, Health Sciences Center, Federal University of Espirito Santo, Vitoria 29075-910, ES, Brazil

**Keywords:** adipogenesis, adipose tissue, monosaturated fatty acids, inflammation, obesity

## Abstract

Adipose tissue (AT) secretes adipokines, modulators of low-grade chronic inflammation in obesity. Molecules that induce the emergence of new and functional adipocytes in AT can alleviate or prevent inflammatory and metabolic disorders. The objective of this study was to investigate the role of palmitoleic acid (n7) in 3T3-L1 and primary pre-adipocyte differentiation and AT inflammation. C57BL/6j mice were submitted to a control or high-fat diet (HFD) for 8 weeks, and treated with n7 for 4 weeks. Mice consuming a HFD presented an increase in body weight, epididymal (Epi) fat mass, and Epi adipocytes size. N7 treatment attenuated the body weight gain and completely prevented the hypertrophy of Epi adipocytes, but not the increment in Epi mass induced by the HFD, suggesting a greater adipocytes hyperplasia in animals treated with n7. It was agreed that n7 increased 3T3-L1 proliferation and differentiation, as well as the expression of genes involved in adipogenesis, such as *Cebpa*, *Pparg*, *aP2*, *Perilipin*, and *Scl2a4*. Furthermore, n7 decreased the inflammatory cytokines *Mcp1*, *Tnfa*, *Il6*, *Cxcl10*, and *Nos2* genes in Epi vascular stromal cells, but not in the whole AT. These findings show that n7 exerts anti-hypertrophic effects in adipocytes which influence the surrounding cells by attenuating the overexpression of pro-inflammatory cytokines triggered by a HFD.

## 1. Introduction

Complications associated with obesity are directly related to white adipose tissue (WAT) expansion and adipo(cyto)kines secretion. In this regard, hypertrophic WAT expansion, as well as the increased expression of pro-inflammatory cytokines, are indicators of dysfunctional WAT and metabolic derangements [1].

Adipocytes and pre-adipocytes of obese WAT show increased expression of monocyte chemotactic protein-1 (MCP-1), a pro-inflammatory cytokine that leads to macrophages infiltration, accumulation, and contributes to inflammation [2]. Therefore, chronic low-grade inflammation is a major hallmark of the obese WAT and predisposes to insulin resistance (IR) leading to type 2 diabetes mellitus (T2 DM). A high-fat diet (HFD) is directly associated with a higher number of M1 (classically activated) macrophages in obese WAT than in the WAT of lean organisms [3]. M1 macrophages are considered key mediators of obesity-induced IR as they increase the expression of pro-inflammatory cytokines, such as tumor necrosis factor-α (TNF-α) and interleukin (IL)-6, and the levels of reactive oxygen and nitrogen intermediates. On the contrary, M2 macrophages protect against obesity-induced AT inflammation and IR by expressing higher levels of anti-inflammatory cytokines such as IL-10 and arginase-1 (Arg-1), which facilitate the maintenance of local immune and metabolic homeostasis [3,4]. The increased ratio of proinflammatory M1 to resident M2 macrophages is a peculiarity of AT inflammation in murine visceral obesity [5] and links metabolic disease to IR [6].

Macrophages are part of the AT-derived stromal vascular fraction (SVF), which is a heterogeneous, versatile, and clinically relevant cell system. The SVF is comprised also of endothelial cells, other immune cells, fibroblasts, smooth muscle cells, mural cells, blood cells, and mesenchymal stem cells (MSCs), also known as adipose-derived stem cells (AdSCs) [7]. Studies have shown functional, immunological, and pathological effects of SVF cells and AdSCs [8,9]. 

Inflammatory conditions affect epigenetics, gene expression and the function of AdSCs [10]. On the other hand, soluble factors secreted by AdSCs appeared to dominate the polarization of macrophages [11], although underlying mechanisms for the crosstalk between AdSCs and macrophages remain to be unveiled. Lastly, some studies also show that cells from the SVF are more effective than AdSCs in mediating important immunomodulatory effects [12]. 

The characteristics of each cell population in the AT and their specific roles in adipocytes (dys) function remain an area of ongoing investigation. 

We have studied the effects of palmitoleic acid (C16:1n7) or n7, an omega-7 monounsaturated long chain fatty acid (LCFA), whose main vegetable sources are macadamia seed oil and sea buckthorn pulp, on mature adipocytes and WAT [13,14]. More recently, we showed that n7 increases the metabolic and oxidative capacity of adipocytes [15]. Additionally, n7 promotes metabolic changes and partially prevents the effects of obesity, mitigating the body mass gain and modulating the expression of genes participating in WAT metabolic pathways [16]. Also described were relevant n7 effects on the pancreas [17], skeletal muscle, and liver [18]. N7 administration improves systemic insulin sensitivity, plasma glucose, plasma lipid profile, and body mass index (BMI) [19,20]. N7 has anti-inflammatory effects on macrophages from the intraperitoneal cavity [21] and bone marrow [22], as well as on J774 macrophages [23]. The effects of n7 on the adipose tissue SVF are unknown.

By combining in vivo and in vitro experiments using visceral AT from obese mice and 3T3-L1 preadipocytes, we demonstrated that palmitoleic acid exerts anti-hypertrophic effects in adipocytes and influences the surrounding stromal cells. N7 attenuates stromal cells’ overexpression of pro-inflammatory cytokines triggered by a HFD-induced obesity associated with low-grade inflammation. This issue is of great relevance considering that obesity and its multiple comorbidities are one of the leading causes of death worldwide.

## 2. Results

### 2.1. Palmitoleic Acid Reduces Body Weight Gain and Adipocyte Hypertrophy in the Epididymal Adipose Tissue of Obese Mice Induced by HFD

Mice receiving HFD showed a decrease in both food (by 50%; Figure 1A) and caloric (by 50%; Figure 1B) intake, but an increase in lipid intake (~3.3×, Figure 1C). Palmitoleic acid did not change these results. Additionally, at the end of the experimental protocol, the HFD and HFD + n7 groups presented a higher body weight than the CO diet group (Figure 1D). However, the body weight gain during the last 4 weeks (n7 treatment period) was lower in the HFD + n7 group (decrease of 33%, compared to HFD; Figure 1E), confirming our previously published results [16].

Mice consuming the HFD presented a significant increase in the mass of the epididymal (Epi) fat depot (by ~2.2×; Figure 1F) and the volume/size of its fat cells. The n7 treatment did not attenuate the Epi mass increase, but completely prevented the hypertrophy of Epi adipocytes (Figure 1G).

#### 2.1.1. Palmitoleic Acid Promotes Proliferation and Differentiation of Preadipocytes

Although palmitoleic acid reversed adipocytes hypertrophy, it did not reverse the increase in visceral (Epi) fat mass induced by the HFD, suggesting that n7 treatment promoted a greater adipocyte hyperplasia. To further investigate this finding, we treated the 3T3-L1 cell line, well characterized and widely used for adipogenesis studies, with vehicle and palmitoleic acid. As a control of the substrate effect of fatty acids, the repercussions of palmitic (a saturated, 16:0) fatty acid in culture cells were also analyzed.

First, we performed proliferation analysis of 3T3-L1 cells treated or not with 100 μM of the fatty acids, for 24 h. Treatment with palmitoleic, but not with palmitic acid, increased the number of cells by about 17% (Figure 2A).

To verify the effect of palmitoleic acid on adipogenesis, 3T3-L1 pre-adipocytes were differentiated for 6 days with a medium containing palmitoleic acid. On the sixth day of differentiation we analyzed adipocytes for lipid content and expression of genes related to adipogenesis and adipocyte maturation. The representative images of Oil red O staining demonstrated that palmitoleic acid increased lipid accumulation in these cells (Figure 2B), which were supported by quantitative data on image analysis (26% increase), and is in agreement with the stimulatory effect observed on the expression of adipogenic markers (Figure 2C–H).

The transcription factors CCAAT/enhancer-binding protein alpha (C/EBPα) and peroxisome proliferator-activated receptor gamma (PPARγ) are the major regulators of early adipogenesis, but are also required to maintain the differentiated state of mature adipocytes and their insulin sensitivity [24]. Palmitoleic acid upregulated C/EBPα and PPARγ mRNA expression in 3T3-L1 adipocytes by 60 and 40%, respectively (Figure 2C,D). In the same way, n7 treatment also increased the mRNA expression of differentiation markers, such as fatty acid binding protein 4, commonly known as adipocyte protein 2 (FABP4/aP2), perilipin, and the glucose transporter 4/Slc2a4 by 60, 200 and 50%, respectively (Figure 2E–G).

#### 2.1.2. HFD Increases the Expression of Genes Related to Inflammation in Whole Epididymal Adipose Tissue, but Palmitoleic Acid Decreases the Expression of Gene Encoding the Inflammatory Cytokine *NOS2*

HFD increased the mRNA expression of *Mcp1* (~5.7×), *Tnfa* (~2.9×), *Il6* (~1.7×) and *Nos2* (~1.8×) encoding proteins related to the pro-inflammatory profile (Figure 3A–C,E). However, we did not observe any considerable difference when comparing the HFD and HFD + n7 groups, except for a reversal of *Nos2* expression (Figure 3E) and an increment in mRNA content that encodes the anti-inflamatory cytokine IL-10 (increase ~2.3×, Figure 3F) in Epi-WAT from animals treated with n7. There were no significant differences in *Cxcl10* (Figure 3D), *Arg1* (Figure 3G), and *Irf4* (Figure 3H) gene expression.

#### 2.1.3. HFD Increases the Expression of Genes Related to Inflammation in Epididymal Vascular Stromal Cells, but Palmitoleic Acid Decreases the Expression of Genes Encoding the Inflammatory Cytokines MCP-1, TNF-α, IL-6, CXCL10, and NOS2

In Figure 4 are illustrated the data obtained by the expression analysis of genes that express the same pro- and anti -inflammatory cytokines in Epi-SVF cells from the animals. Regarding the genes encoding pro-inflammatory proteins related to polarized (M1) macrophage markers, we observed an increase in the expression of *Mcp1* (~3.5×), *Tnfa* (~1.8×), *Il6* (~1.7×), Cxcl10 (~2×), and *Nos2* (~1.5×) in the obese group compared to the control group, but this increase was not observed in the obese group + n7 (Figure 4A–E), showing that palmitoleic acid was able to treat the inflammation triggered by HFD The analysis of genes encoding anti-inflammatory markers and non-polarized macrophages (M2) did not show a statistically relevant difference between the groups (Figure 4F–H).

#### 2.1.4. Palmitoleic Acid Does Not Change the Basal Expression of Genes Related to Inflammation in 3T3-L1 Adipocytes

To investigate whether palmitoleic acid regulates pro-inflammatory genes expression in adipocytes, differentiated 3T3-L1 cells were treated with palmitoleic (or palmitic) fatty acids. As depicted in Figure 5, treatment with the saturated palmitic acid induced a significant increase in both *Mcp1* (by 50%) and *Tnfa* (by 80%) mRNA content in comparison to those treated with vehicle, although it did not alter *Il6, Cxcl10, Il10,* and *Arg1* genes expression. N7 treatment (per se) did not affect the expression of any of the cytokines genes analyzed (Figure 5A–F).

## 3. Discussion

Recent studies indicate that palmitoleic acid increases insulin sensitivity, improves blood lipid profile, alters macrophage differentiation in rodent models and in cell culture, and has a potential impact on certain metabolic diseases [21,25]. 

To better understand the mechanisms underlying palmitoleic acid anti-inflammatory and anti-hypertrophic effects in WAT we investigated the effects of n7 in (adipose)-derived SVF cells and tissue from a dysfunctional visceral fat depot triggered by HFD-induced obesity/low-grade inflammation and also in the 3T3-L1 cell line.

Our results showed that there was an increase in body mass in obese animals induced by the HFD and that palmitoleic acid prevented the body mass gain, as well as reduced adipocyte hypertrophy, but was not able to reverse the increase in Epi-WAT depot mass. These data suggested greater hyperplasia in the WAT of animals treated with n7. To further investigate, we treated the 3T3-L1 cell line with both C16:1n7 or C16:0 fatty acids. N7, but not the saturated (C16:0) palmitic acid, increased the proliferation and number of differentiated cells, with visualization and quantification of a greater amount of lipids in their interior. Furthermore, n7 increased the expression of the transcription factors *Ppar-γ2* and *Cebpa*, which are considered the master regulators of adipogenesis, and when activated, induce the expression of target genes which encodes the proteins necessary for the maintenance of the adipocyte phenotype [26,27].

Corroborating these results, we also detected an increase in the expression of main terminal differentiation markers of adipocytes that respond to *Ppar-γ* and *Cebpa*: 1. *Fabp4*, which encodes the fatty acid binding protein (aP2); 2. *Plin1*, which encodes perilipin A, a protein present in the lipid droplets, with an important role in regulating lipolytic activity; 3. *Glut4*, which encodes the protein responsible for the transport of glucose in response to insulin; and 4. *AdipoQ*, which encodes adiponectin, an adipokine with insulin-sensitizing effects. Thus, our results suggest that n7 induces the formation of new functional and healthy adipocytes.

It was suggested that obesity induces adipocyte hypertrophy, increases the metabolic rate (such as lipolysis), hypoxia and oxidative stress in visceral adipose tissue [28]. Adipocyte hypertrophy curses with immune cell infiltration and increased expression of pro-inflammatory cytokines, and adipose tissue dysfunction [29]. On the other hand, it was also suggested that in obesity “healthy” WAT expansion is achieved by recruiting and differentiating adipose precursor cells rather than inflating fat into mature adipocytes. In metabolically unhealthy obesity, the storage capacity of subcutaneous adipose tissue, the largest WAT depot, is limited. Further caloric overload due to obesity leads to fat accumulation in ectopic tissues and in the visceral adipose depots, an event commonly defined as “lipotoxicity” [30].

Since our results suggest the presence of functional (not hypertrophied) adipocytes in WAT from obese mice treated with palmitoleic acid, we next investigated its effect on treating/ reversing the pro-inflammatory environment in WAT, a chronic state associated with HFD-induced obesity, that blocks adipocyte insulin action and leads to the development of IR and 2DM. 

Looking at Epi-(whole) WAT, our data showed that HFD induced an increase in *Mcp1*, *Tnfa*, *Il6* and *Nos2* mRNA content, but we did not observe any considerable differences when comparing the HFD and HFD + n7 group, except for a reversal of *Nos2* and an increment in Il10 mRNA levels in animals treated with nN7. However, when we investigated the action of n7 in Epi WAT-derived SVF cells, relevant effects of this monounsaturated FA were observed, since the increase in gene expression of *Mcp1*, *Tnfa*, *Il6*, *Cxcl10* and *Nos2* induced by HFD in the visceral SVF cells was completely reversed by the treatment.

It is known that MCP-1 is a crucial mediator of chronic inflammation in visceral AT. Local and systemic levels of MCP-1 are increased in obese mice and humans [31] and can occur as early as 2 days post initiation of a high-fat diet (HFD) in murine visceral AT [32]. In sustained obesity, M1 macrophages become the main producers of MCP-1 and provide a positive feedback signal to recruit additional M1 macrophages [33]. However, M1 macrophages are not present in large numbers in murine adipose tissue until at least 8 weeks of a HFD [34], suggesting that preadipocytes and resident stromal cells in the SVF are responsible for early MCP-1 secretion and macrophage accumulation [35,36]. Moreover, using flow cytometry to quantify the number of immune cell populations in the Epi depot, a proportion of only 15% of immune cells in the total SVF was found in male mice at 8 months of age [37].

M2 macrophages are involved in immunoregulation/tissue repair and homoeostasis, producing IL-10, an anti-inflammatory cytokine, which may protect against inflammation. Thus, restoring M2 macrophages in obese individuals by various approaches contributes to the resolution of associated inflammation and IR [5,6]. Furthermore, activation of GPR120 in the macrophage lineage RAW264.7 decreased *Tnfa* expression [38]. Interestingly, GPR120 has been described to act as a receptor for palmitoleic acid in both AdSCs and adipocytes [38,39]. Thus, one hypothesis is that n7 acts at least in part, via GPR120 in both AdSCs and macrophages to promote adipogenesis (contributing to the greater hyperplasia and lesser hypertrophy observed) and to exert anti-inflammatory effects, respectively. Taken together, these effects culminate in the beneficial results described here for n7-treated obese mice.

Given the above, as the stromal fraction is composed of several cells of a highly heterogeneous nature, including pre-adipocytes, macrophages, and AdSCs, and has protective characteristics, this cell fraction plays an essential role in controlling the inflammation associated with obesity and metabolic disorders [11,40]. Our data demonstrated a higher expression of IL10 in the epididymal stromal fraction of HFD + n7 animals, suggesting that palmitoleic acid may provide a beneficial effect by increasing anti-inflammatory cytokine expression. This could provide the adipose tissue with a supplementary mechanism to maintain the balance between the pro-inflammatory and anti-inflammatory cytokines expression despite the inflammatory signals triggered by a HFD.

The n7 anti-inflammatory and anti-hypertrophic effects in WAT, associated with greater adipocytes hyperplasia shown in this work, seem to correlate with its effect on decreasing non-alcoholic hepatic steatosis and increasing lipogenesis and fatty acid oxidation in the WAT of obese mice [16]. Young healthy adipocytes show a higher lipogenesis to lipolysis ratio and store the excess fatty acids and cholesterol provided by a HFD preventing lipid accumulation in ectopic sites and plasma cholesterol increase.

In summary, after 8 weeks of HFD-induced obesity in mice, we observed that palmitoleic acid had a beneficial effect on the changes triggered by the HFD, attenuating the body mass gain and reversing adipocyte hypertrophy. Furthermore, it decreased the expression of pro-inflammatory genes in the SVF and increased the proliferation and differentiation of 3T3-11 cells. Herein, we showed that n7 modulates the functional capabilities of stromal vascular fraction cells during the process of inflammation induced by HFD, and our data reinforce that there is a crosstalk between adipocytes and adipose-derived stromal cells to promote n7 anti-hypertrophic and anti-inflammatory synergistic effects in obese mice. These results elucidate new agent and cellular targets to promote and modulate the healthy expansion of WAT. Forward, studies may clarify mechanisms that palmitoleic acid n7 uses for metabolic protection and the reduction of inflammation in WAT.

## 4. Materials and Methods

### 4.1. Animals, Diets, and Palmitoleic Acid Supplementation

All procedures were approved by the Ethics Committee on Animal Use of the Federal University of São Paulo (CEUA nº 8827141217). Eight-week-old male C57Bl/6J mice were obtained from the Center for Development of Experimental Models (CEDEME), Federal University of São Paulo, and maintained under controlled light-dark cycle of 12/12 h, temperature of 22 ± 1 °C, and relative humidity 53 ± 2%, with free access to feed and filtered water. The experimental protocol remained for 8 weeks. In the first 4 weeks (period I), mice were divided into two groups: Control and high-fat diet (HFD) groups, both prepared according to the recommendations of the American Institute of Nutrition (AIN-93) for adult mice [41], differing only in energy density (CO = 3803 kcal/kg, HFD = 5340 kcal/kg diet). Lard [which contains 40% saturated FAs (24% palmitic and 44% oleic acid)] and soybean oil [which contains 81% unsaturated (24% oleic and 54% linoleic acid)] were used as a source of lipids. In the next 4 weeks (period II) the HFD group was subdivided into HFD and HFD supplemented with palmitoleic acid (HFD + n7) groups. A control diet contains 76% carbohydrate, 15% protein and 9% fat and a HFD contains 26% carbohydrate, 15% protein and 59% fat, in % kcal, as we detailed before [42]. Supplementation was performed by oral gavage at 300 mg/kg/day (~10 μL, according to the body weight) of pure palmitoleic acid [Sigma, St. Louis, MO, USA] [24] or water. 

### 4.2. Experimental Procedure

Body weight and food intake were measured 3 times a week. After 8 weeks of the experimental protocol, 10–12 h fasted mice were anesthetized with isoflurane and sacrificed by cervical dislocation, Epididymal (Epi) WAT was excised, weighed and processed as described below.

#### 4.2.1. Adipocytes and Stromal Vascular Fraction (SVF)

Epididymal fat depot was removed and placed in a digestion buffer. Samples were minced and digested using collagenase as previously described [43]. Briefly, fine fragments were incubated in a digestion buffer (Dulbecco’s modified Eagle’s medium-D’MEM/HEPES 20 mM/BSA 4%, *collagenase II* [Sigma Chemical, St. Louis, MO, USA] −1.0 mg/mL, pH 7.40) for ~45 min at 37 °C in a bath shaker with orbital agitation (150 rpm). The homogenate was filtered through nylon mesh (Corning, NY, USA) and centrifuged (400× *g*, 1 min), and then divided into two fractions: 1. supernatant, which contains the isolated mature adipocytes; these were washed 3 times in a fresh buffer without collagenase, and adipocytes were harvested for morphometric analysis. Aliquots of cell suspension were placed in a microscope slide and 6 fields were photographed under an optical microscope (×100 magnification) coupled to a camera (AxioCam ERc5s; Zeiss, Oberkochen, Germany), and the mean adipocyte volume (4/3 × π × r3) was determined by measuring 50 cells using AxioVision LE64 software. 2. All remaining filtrates containing the SVF, which was subjected to centrifugation (1500× *g* for 10 min) to form a cellular pellet, were washed twice and aspirated. Then, the SVF was incubated on ice for 10 min with red cell lysis buffer (Roche Diagnostics GmbH, Mannheim, Germany). The SVF was washed with PBS containing antibiotics and, once again, centrifuged. Trizol reagent was added and stored in a freezer −80 °C. The SVF was pooled from two mice. One pooled cell was counted as one sample. 

#### 4.2.2. Differentiation of Pre-Adipocytes from the 3T3-L1 Cell Line

Preadipocytes (Swiss 3T3-L1 cells), obtained from the cell bank of Rio de Janeiro (RJ, Brazil) were also cultivated and maintained in D’MEM supplemented with 10% of calf serum (CS) and antibiotics, and kept in a 5% CO2, 37 °C until they reached the confluence (day 2). Cells were induced to differentiation on day 0 by the addition of an adipogenic cocktail containing 0.5 mM IBMX (3-isobutyl-1-methylxanthine), 1 µM dexamethasone, 1.67 µM insulin, and D’MEM supplemented with 10% of fetal bovine serum (plus antibiotics). After 48 h, the medium was changed to the same D’MEM (10% FBS) containing 0.4 µM insulin (maintenance medium), which was renewed every 2 days [44] until day 6.

#### 4.2.3. Treatment with Fatty Acids

Palmitic acid (PA) or palmitoleic acid (n7) (Sigma Chemical, St. Louis, MO, USA) at a concentration of 100 µM, was dissolved in ethanol (vehicle) not exceeding 0.05% and added to the cells during the proliferation (MTT assays) on day 0 with the differentiation induction medium and maintained until the end of the respective assay. Fatty acids concentration was pre-defined by cytotoxicity tests [15]. 

#### 4.2.4. Cell Proliferation Assay

Proliferation of 3T3-L1 preadipocytes was evaluated by the 3-[4,5-dimethylthiazol-2-diphenyltetrazolio] (MTT) formazan bromide reduction method by the MTT cell proliferation Kit (Cat. No. 11465007001, Roche Diagnostics, Mannheim, Germany) [45]. After 24 h of culture (5 × 10^3^ cells/well with 100 μL D’MEM/CS) in 96-wells plates (flat bottom) in the presence or absence of the respective FA, 100 μL/well of MTT were added to cells and incubated for 4 h (37 °C, 5% CO_2_). After this period, a formazan crystals solubilization solution (10% SDS at HCl 0.01 M) was added at 10 μL/well. The plates were then incubated for 14 h (37 °C, 5% CO_2_). After incubation, the absorbance (at 550 nm) was measured in an optical plate reader. Since all cultures were plated with the same initial number of cells, an increase or decrease in viable cells represents the potential for proliferation of pre-adipocytes, which were expressed as percentage values compared to the vehicle.

#### 4.2.5. Oil Red O Staining and Lipid Content Determination

After 6 days of differentiation induction, 3t3-11 adipocytes were washed with PBS and fixed with 10% formalin in PBS, and stained in 0.3% solution of red oil (Sigma) at 60% (*v*/*v*) 2-propanol in water (followed by filtration) for 1 h, for marking, visualization and quantification of lipid content as described by [46]. Images were obtained with a phase-contrast microscope at low magnification (20×), and the lipid content was analyzed by spectrophotometry.

#### 4.2.6. RNA Extraction and Quantitative Real-Time Polymerase

The total RNA was extracted from both the Epi adipose-derived SFV and the whole adipose depot, and also from the 3t3-11 cell lysate, using Trizol reagent (Invitrogen Life Technologies, Waltham, MA, USA). The total RNA was analyzed for quality on ratios 260/280 and 260/230 nm on NANODROP (Thermo Scientific, Waltham, MA USA), and reverse transcribed to cDNA using the Superscript III cDNA kit (Thermo Scientific, EUA, Waltham, MA USA). Gene expression was assessed by chain reaction quantitative real-time polymerase (PCR) using a Rotor gene (Qiagen) and fluorescent dye such as SYBR Green as previously described [24]. Analysis of real-time PCR data was performed using the 2^−ΔΔCT^ method. Data are expressed as the ratio of target gene expression to housekeeping gene (*Gapdh* and *36b4*). The primers used are shown in Table 1.

### 4.3. Statistical Analysis

Data were analyzed by one-way analysis of variance (ANOVA), followed by Tukey’s post-test for comparison between groups. Student’s t test was used for body weight gain, and expressed as mean ± standard error of the average (SEM). Differences were considered significant for *p* < 0.05. Statistical analysis was performed using a GraphPad Prism software version 9.1.2 (GraphPad Software Inc., San Diego, CA, USA).

## Figures and Tables

**Figure 1 pharmaceuticals-15-01194-f001:**
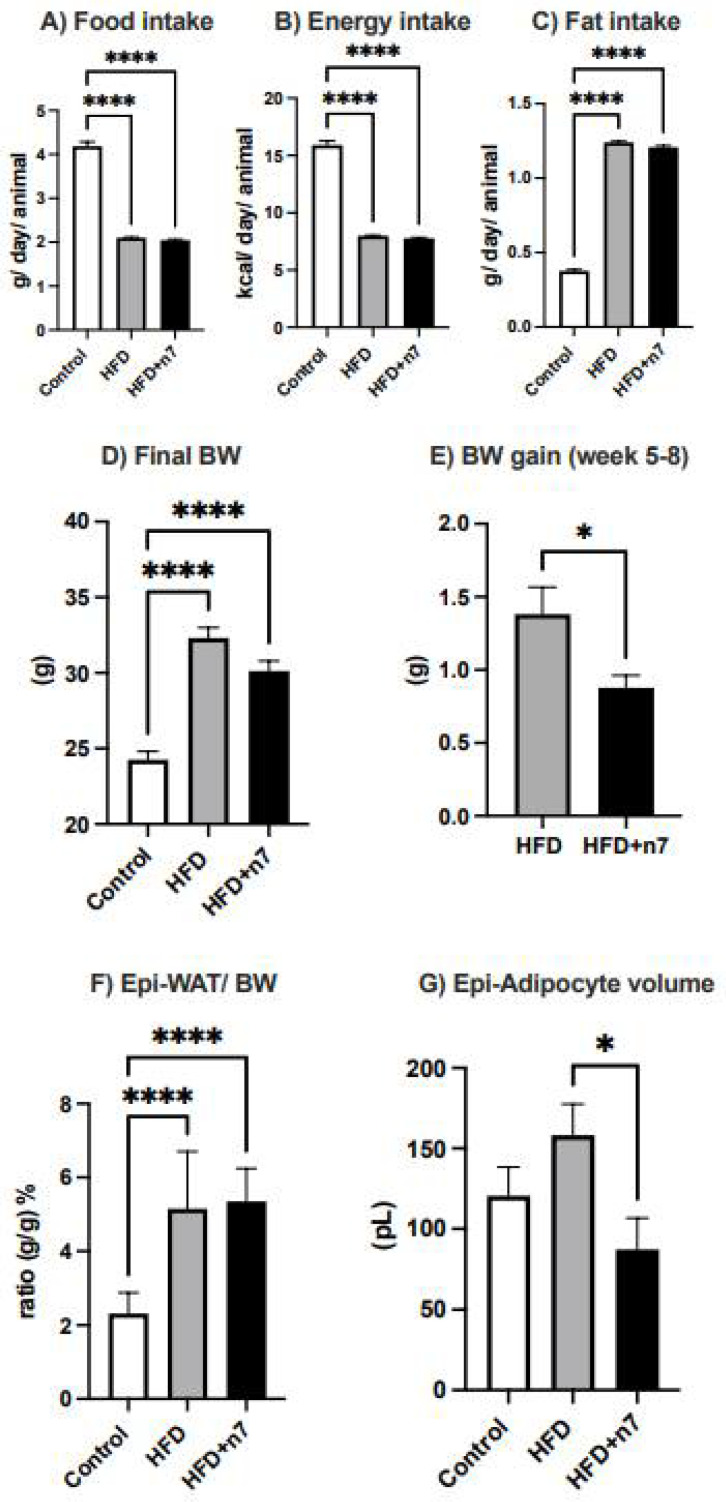
Effects of high fat diet (HFD) and palmitoleic acid (N7) supplementation on food consumption, body weight gain, adiposity and hypertrophy of mice adipocytes. Food (**A**), energy (**B**), and fat (**C**) intake. At the weeks 5–8, the diets were maintained and mice received water (control and HFD groups) or palmitoleic acid (HFD + n7 group) by oral gavage. Final body weight (BW) (**D**), BW gain during weeks 5 to 8 (**E**), relative epididymal (Epi)-WAT mass per BW (**F**) and Epi-adipocytes volume (**G**). Values are mean ± SEM (n = 12). * *p* < 0.05 or **** *p* < 0.0001 vs. Control.

**Figure 2 pharmaceuticals-15-01194-f002:**
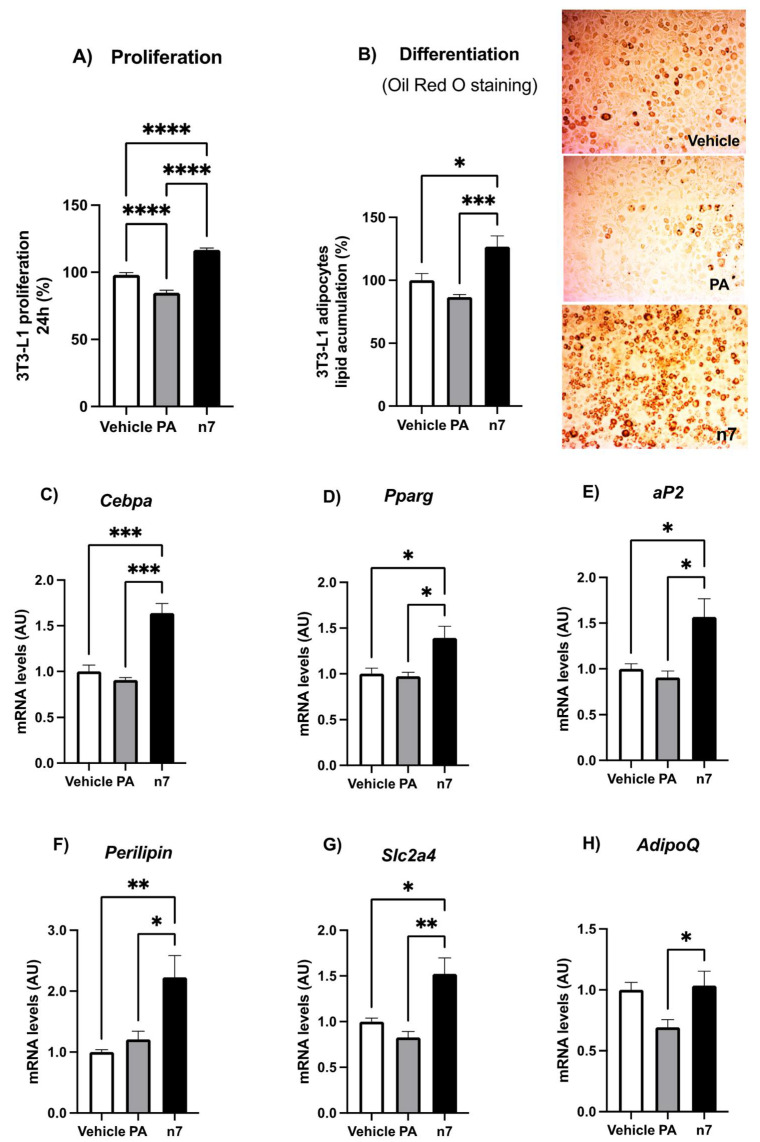
Effects of palmitoleic acid on 3T3-L1 cell proliferation and differentiation. (**A**) Preadipocytes proliferation determined by MTT assay. Cells were plated into 96-well plates at a density of 3 × 10^3^ cells/100 μL per well. These cells were then cultured in media with vehicle, palmitic (PA, 100 µM) or palmitoleic acid (n7, 100 µM) for 24 h. (**B**) Lipid accumulation determined by oil red O staining and spectrophotometry. Images were obtained with a phase-contrast microscope at low magnification −20×. mRNA levels of adipogenic markers: *Cebpa* (**C**), *Pparg* (**D**), *aP2* (**E**), *Perilipin* (**F**), *Scl2a4* (**G**) and *AdipoQ* (**H**). Values of mRNA were expressed as arbitrary units (AU) in relation to the control and corrected by the expression of the constitutive gene *36B4*. In B-H, experiments were performed in differentiated 3T3-L1 adipocytes (day 6) treated with the fatty acids on day 0 (with the differentiation induction medium) and maintained until the end. Values are mean ± SEM of six samples from two independent experiments. * *p* < 0.05, ** *p* < 0.001, *** *p* < 0.0001 or **** *p* < 0.00001 vs. Control.

**Figure 3 pharmaceuticals-15-01194-f003:**
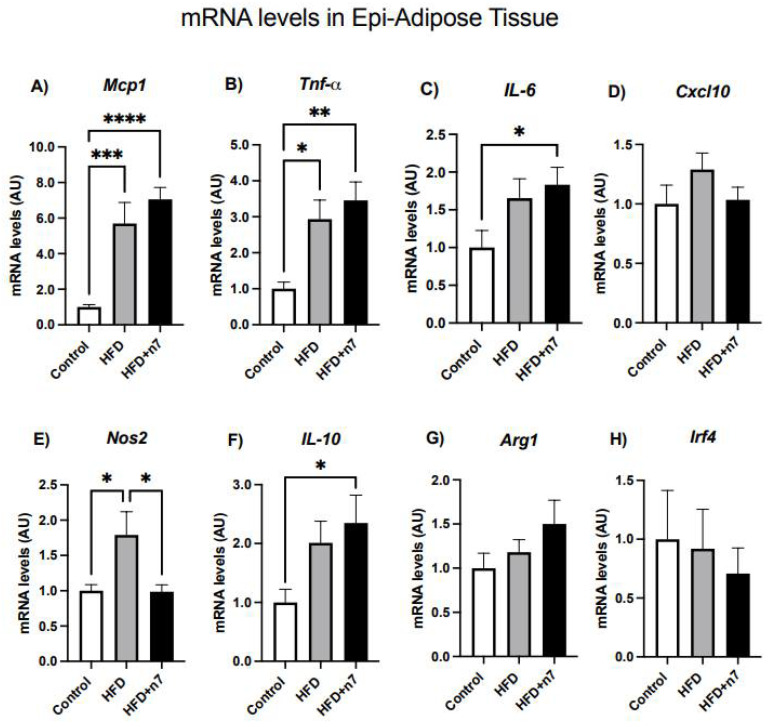
Effects of HFD and palmitoleic acid supplementation on mRNA levels of cytokines expressed by Epi-WAT. (**A**) *Mcp1*, (**B**) *Tnfa*, (**C**) *Il6*, (**D**) *Cxcl10*, (**E**) *Nos2*, (**F**) *Il10*, (**G**) *Arg1*, and (**H**) *Irf4.* Values of mRNA were expressed as arbitrary units (AU) in relation to the control and corrected by the expression of the constitutive gene *Gapdh*. Values are mean ± SEM (n = 12). * *p* < 0.05, ** *p* < 0.001, *** *p* < 0.0001 or **** *p* < 0.00001 vs. Control.

**Figure 4 pharmaceuticals-15-01194-f004:**
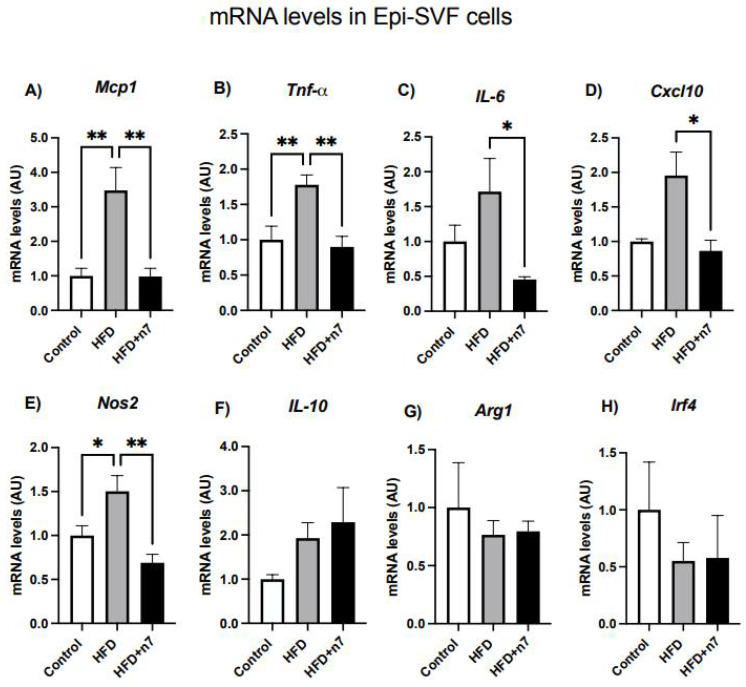
Effects of HFD and palmitoleic acid supplementation on mRNA levels of cytokines expressed by Epi-SVF cells. (**A**) *Mcp1*, (**B**) *Tnfa*, (**C**) *Il6*, (**D**) *Cxcl10*, (**E**) *Nos2*, (**F**) *Il10*, (**G**) *Arg1* and (**H**) *Irf4.* Values of mRNA were expressed as arbitrary units (AU) in relation to the control and corrected by the expression of the constitutive gene *Gapdh*. Values are mean ± SEM (n = 4, SVF cells were pooled from three mice). * *p* < 0.05 or ** *p* < 0.001 vs. Control.

**Figure 5 pharmaceuticals-15-01194-f005:**
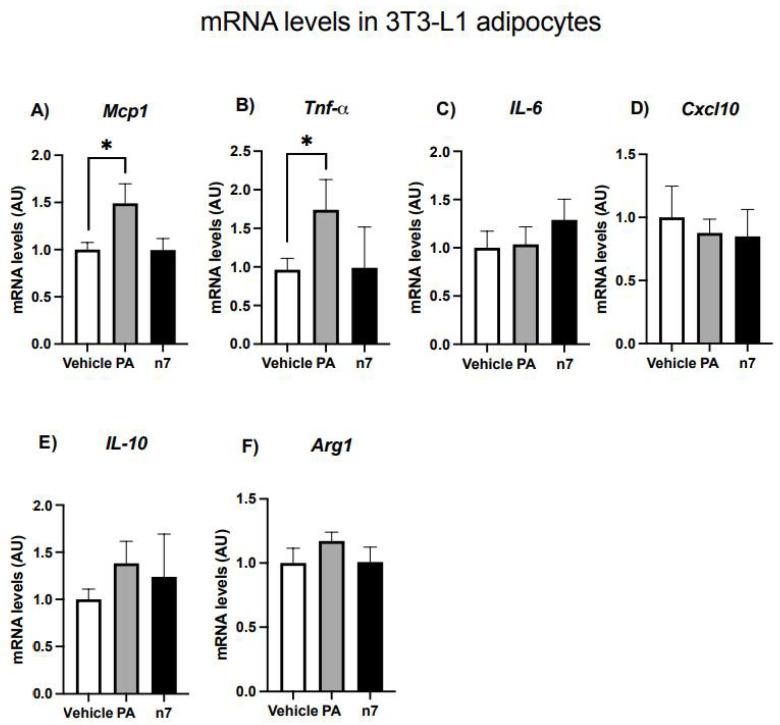
Effects of palmitoleic acid on mRNA levels of cytokines expressed by 3T3-L1 adipocytes. (**A**) *Mcp1*, (**B**) *Tnfa*, (**C**) *Il6*, (**D**) *Cxcl10*, (**E**) *Il10* and (**F**) *Arg1.* Values of mRNA were expressed as arbitrary units (AU) in relation to the control and corrected by the expression of the constitutive gene *36B4*. Experiments were performed in differentiated 3T3-L1 adipocytes (day 6) treated with the fatty acids on day 0 (with the differentiation induction medium) and maintained until the end. Values are mean ± SEM of six samples from two independent experiments. * *p* < 0.05 vs.Vehicle.

**Table 1 pharmaceuticals-15-01194-t001:** Sense and antisense sequences of primers used in real-time RT-PCR.

Gene	5′ Primer (5′-3′)-Sense	3′ Primer (5′-3′)-Antisense	Annealing
*AdipoQ*	GCAGAGATGGCACTCCTGGA	CCCTTCAGCTCCTGTCATTCC	60 °C
*Arg1*	GCACTCATGGAAGTACACGAGGAC	CCAACCCAGTGATCTTGACTGA	60 °C
*Cebpα*	CGCAAGAGCCGAGATAAAGC	CAGTTCACGGCTCAGCTGTTC	60 °C
*Cxcl10*	GACGGTCCGCTGCAACTG	GCTTCCCTATGGCCCTCATT	60 °C
*Fabp4/Ap2*	AAGGTGAAGAGCATCATAACCCT	TCACGCCTTTCATAACACATTCC	60 °C
*Gapdh*	CCACCACCCTGTTGCTGTAG	CTTGGGCTACACTGAGGACC	60 °C
*Slc2a4*	CATTCCCTGGTTCATTGTGG	GAAGACGTAAGGACCCATAGC	60 °C
*Irf4*	CAAAGCACAGAGTCACCTGG	TGCAAGCTCTTTGACACACA	60 °C
*Il6*	TTCTCTGGGAAATCGTGGAAA	TCAGAATTGCCATTGCACAAC	60 °C
*Il10*	CTGGACAACATACTGCTAACCG	GGGCATCACTTCTACCAGGTAA	60 °C
*Mcp1*	GCCCCACTCACCTGCTGCTACT	GCCCCACTCACCTGCTGCTACT	60 °C
*Nos2*	GCCACCAACAATGGCAACA	CGTACCGGATGAGCTGTGAATT	60 °C
*Pparγ2*	GCATCAGGCTTCCACTATGGA	AAGGCACTTCTGAAACCGACA	60 °C
*Plin*	AGTGTGGGGTCCTTGGGCGT	TGGCAGCTGTGAACTGGGTGG	60 °C
*Tnfa*	CCCTCACACTCAGATCATCTTCT	GCTACGACGTGGGCTACAG	60 °C
*36b4*	TAAAGACTGGAGACAAGGTG	GTGTACTCAGTCTCCACAGA	60 °C

*AdipoQ*, adiponectin; *Arg1*, arginase, liver; *Cebpa* (C/EBP-α), CCAAT/enhancer-binding protein, alpha; *Cxcl10*, chemokine (C-X-C motif) ligand 10; *Fabp4*, fatty Acid Binding Protein 4; *Gapdh*, glyceraldehyde-3-phosphate Dehydrogenase; *Slc2a4* (Glut-4), glucose transporter 4; *Irf4*, interferon regulatory factor 4; *I**l6*, interleukin 6; *I**l10*, interleukin 10; *Mcp1/Ccl2*, monocyte chemoattractant protein 1; *Nos2*, nitric oxide synthase 2; *Pparγ2*, peroxisome proliferator activated receptor gamma 2; *Plin*, perilipin 1; *Tnfa*, tumor necrosis factor alpha; *36b4*, Acidic ribosomal phosphoprotein P0 (RPL0).

## Data Availability

Data is contained within the article.

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
