# Peer review of "Palmitoleic Acid Acts on Adipose-Derived Stromal Cells and Promotes Anti-Hypertrophic and Anti-Inflammatory Effects in Obese Mice"

_pharmaceuticals, 2022, doi:10.3390/ph15101194_

Round 1
Reviewer 1 Report
Manuscript ID: pharmaceuticals-1898151
Title: Palmitoleic acid acts on adipose-derived stromal cells and enxerts
anti-hypertrophic and anti-inflammatory effects in obese mice
Overall:
† These authors contributed equally to this work and share last authorship “last OR first???”
Why the parameters of hypercholesterolemia were not measured to show the generation of hypercholesterolemic models? And why the histopathological investigations of different groups were not included?
The authors should explain why the parameters indicating the existence of HFD-induced oxidative stress or insulin resistance were not included.
The English language should be improved; there is a spelling mistake in the title.
Regarding the title, did the anti-hypertrophic and anti-inflammatory activities of Palmitoleic acid linked to its hypocholesterolemic potential? The authors should highlight this connection in the section “Discussion”.
Abstract:
This section should be improved; the experiments should be added according to their order of appearance (in vivo followed by in vitro). The authors should define the word of “3T3-L1” at its first mention.
Please, include genes involved in adipogenesis as those involved in inflammation.
Keywords:
Why “MUFA” is included? It is mentioned once in the whole manuscript.
Materials and methods:
The authors should provide more details on the preparation of HFD, supported with references.
The title should be “Adipocytes and stromal vascular fraction (SVF)” instead of “Adipocytes and SVF Isolation”.
On page 13, line 350: the ethics protocol should be mentioned once in the first paragraph of “animals”.
Results:
It would be more representative to present Figure 1E in line graph for each group during the whole experimental period.
The vertical lines of genetic expression figures should have a title of “mRNA levels (AU)” in all figures.
The authors should reduce the use of “As expected”
Unify the word “TNF-α” and “N7”
Please, provide your responses in a list of points.
Author Response
Overall:
- † These authors contributed equally to this work and share last authorship “last OR first???”
Response: First.
We apologize for our mistake and thanks the Reviewer for alerting us. We have now adjusted accordingly(highlighted in red).
- Why the parameters of hypercholesterolemia were not measured to show the generation of hypercholesterolemic models? And why the histopathological investigations of different groups were not included?
Response: We understandthe relevant comment pointed out by the Reviewer. Anyway, using the same experimental model used in the present study, we have already shown that animals submitted to a high fat diet show an increase in plasma triglycerides, total cholesterol and fractions (Nutrients. 2021 Feb 15;13(2):622. doi: 10.3390/nu13020622; Nutrients. 2021 Feb 26;13(3):754. doi: 10.3390/nu13030754; Physiol Rep. 2020 Feb;8(4):e14380. doi: 10.14814/phy2.14380; Front Endocrinol. 201 Nov 5;10:750. doi: 10.3389/fendo.2019.00750; Cells. 2019 Sep 6;8(9):1041. doi: 10.3390/cells8091041), however, when supplemented with palmitoleic acid, we observed that it was not able to reversing these parameters in plasma dosages, but it prevented hepatic stetosis in obese mice (Front Endocrinol. 2020 Sep 30;11:537061. doi: 10.3389/fendo.2020.537061),
Regarding histopathological investigations,due to the impossibility of continuing the experiments that were in progress as a result of events related to COVID-19, we were unable to carry out some experiments that were requested by the Reviewer. However, our group has already demonstrated in histological liver sections (Nutrients. 2021 Feb 15;13(2):622. doi: 10.3390/nu13020622) and images of fat cells (Cells. 2019 Sep 6;8(9):1041. doi: 10.3390/cells8091041) of mice, the deleterious effects of the high fat diet.
- The authors should explain why the parameters indicating the existence of HFD-induced oxidative stress or insulin resistance were not included.
Response: We understand the relevant comment pointed out by the Reviewer. Anyway, as mentioned above, we have already published it before (Front Endocrinol. 2020 Sep 30;11:537061. doi: 10.3389/fendo.2020.537061; J Physiol. 2016 Nov 1;594(21):6301-6317. doi: 10.1113/JP272541), and we have now included this important information in the revised text.
- The English language should be improved; there is a spelling mistake in the title.
Regarding the title, did the anti-hypertrophic and anti-inflammatory activities of Palmitoleic acid linked to its hypocholesterolemic potential? The authors should highlight this connection in the section “Discussion”.
Response: We thank the Reviewer for advising us on the English review necessary to improve the readability of the article. A native English professional has revised the text in the new version of the manuscript. The word “enxerts” was changed by “exerts” in the title.
We also did the connection suggested in the section “Discussion”(see highlighted in red text).
Abstract:
- This section should be improved; the experiments should be added according to their order of appearance (in vivo followed by in vitro). The authors should define the word of “3T3-L1” at its first mention.
Please, include genes involved in adipogenesis as those involved in inflammation.
Response: We thank for all suggestions and correction that are very appropriate. The adjustments were made as suggested by the Reviewer(see highlighted in red in the abstract)
Keywords:
- Why “MUFA” is included? It is mentioned once in the whole manuscript.
Response: We thank the reviewer for pointing this out. “MUFA” has been removed from keywords.
Materials and methods:
- The authors should provide more details on the preparation of HFD, supported with references.
Response:The diets were prepared in our laboratory, based on the AIN-93 (Reeves et al., 1993). For the preparation of HF diet, lard [which contains 40% saturated FAs (24% palmitic and 44% oleic acid)] and soybean oil [which contains 81% unsaturated (24% oleic and 54% linoleic acid)] were used as a source of lipids (9:1 ratio), which represents 59% of the energy in the diet, as we detailed before [de Sá et al., 2016, doi:10.1113/JP272541. We have now amended de text and added the citation in subsection “Animals, Diets, and Palmitoleic Acid Supplementation” (Materials and Methods, highlighted in red text).
- The title should be “Adipocytes and stromal vascular fraction (SVF)” instead of “Adipocytes and SVF Isolation”.
Response: We thank for the suggestion, which is very appropriate. The subtitle was now changed.
- On page 13, line 350: the ethics protocol should be mentioned once in the first paragraph of “animals”.
Response: done.
Results:
- It would be more representative to present Figure 1E in line graph for each group during the whole experimental period.
Response: done
- The vertical lines of genetic expression figures should have a title of “mRNA levels (AU)” in all figures.
Response: done
- The authors should reduce the use of “As expected”
Response: done
- Unify the word “TNF-α” and “N7”
Response: done (Tnfa was used for gene; TNF-α, for protein)
- Please, provide your responses in a list of points.
Response: We thank the Reviewer for pointing out several important comments and suggestions. We are confident that the quality and readability of the manuscript have been substantially improved.

Reviewer 2 Report
Please see the attached file.

Author Response
Reviewer 2
Please see the attached file.
Response: We thank the reviewer for pointing out several important comments in the manuscript.
The text was revised in order to address all details, suggestions and changes pointed out by thereviewer and we have now rewritten and incorporated all, as requested (highlighted in red text).
Some points we would like to address:
- Line 34: Introduction part is too loose, may be more concise
The adjustments were made as suggested by the Reviewer.
- Line 42: T2 DM: changed
- Figure 1: Compared HFD with HFD plus n7, only BW gain (5-8 wks) and Epi-adipocyte volume were observed the difference (p<0.05).
Response: The decrease in adipocyte volume without statistical change in adipose mass suggests the emergence of new adipocytes in the tissue. Data in the literature show that the expansion of adipose tissue by hyperplasia (increment in adipogenesis), improves metabolic diseases caused by obesity (Cell Metab. 2021 doi: 10.1016/j.cmet.2021.05.014; Front Cardiovasc Med. 2020 doi: 10.3389/fcvm.2020.00022), in addition to preventing complications due to the important plasticity of adipose tissue (Nat Med. 2013 doi:10.1038/nm.3324)
- Figure 2: AdipoQ, PA decreased
Response: Saturated fatty acids, including 16:0, are known for their deleterious effects on both metabolism and glucose uptake and insulin signaling in muscle and adipose tissue (Sá et al., 2016; 2020; Bolsoni-Lopes et al., 2014; Palomer et al., 2018; Nardi et al., 2014, Talbot et al., 2014). The mechanisms involved in this process seem to involve inflammatory pathways and increased oxidative stress, such as ROS\IKKb signaling, and reduction in leptin and adiponectin (and/or its receptor) gene expression (Ceja-Galicia et al., 2022; Zhao et al., 2016). On the other hand, data demonstrate that mono- and polyunsaturated fatty acids not only prevent, but also significantly reduce insulin resistance promoted by saturated fatty acids, improving energy metabolism in liver, muscle, and adipose tissue (Nardi et al., 2014; Cao et al., 2008; Bolsoni-Lopes 2013; Cruz et al., 2020). In isolated adipocytes, 16:0 reduces glucose uptake in the presence of insulin, but 16:1 n7 promotes increased glucose uptake, GLUT4 protein content and glut-4 translocation via AMPK, in addition to inducing the oxidation of glucose and fatty acids, making the adipose cell metabolically more active (Cruz et al., 2018; Cruz et al., 2020; Bolsoni-Lopes et L., 2014).
References:
Bolsoni-Lopes et L., 2014: Lipids Health Dis. 2014 doi: 10.1186/1476-511X-13-199
Bolsoni-Lopes et L., 2013: Am J Physiol Endocrinol Metab. 2013 Nov 1;305(9):E1093-102. doi: 10.1152/ajpendo.00082.2013.
Cao et al., 2008: Cell2008 doi: 10.1016/j.cell.2008.07.048
Ceja-Galicia et al., 2022: Life Sci.2022 doi: 10.1016/j.lfs.2021.120262
Cruz et al., 2020: Front Endocrinol 2020 doi: 10.3389/fendo.2020.537061
Cruz et al., 2018: Lipids Health Dis. 2018 doi: 10.1186/s12944-018-0710-z
Nardi et al., 2014: Plos one 2014 doi.org/10.1371/journal.pone.0092255
Palomer et al., 2018: Trends Endocrinol Metab. 2018 doi: 10.1016/j.tem.2017.11.009
Sá et al., 2020: Physiol Rep. 2020 doi: 10.14814/phy2.14380
Sá et al., 2016: J Physiol. 2016 doi: 10.1113/JP272541
Talbot et al., 2014: Molecular and Cellular Endocrinology 2014 doi.org/10.1016/j.mce.2014.06.010
- Figure 3: Compared HFD with HFD plus n7, only Nos2 was found difference (p<0.05). When compared to the figure 4, what is reason. ???
Response: Herein, we believe that the reviewer is referring to figure 3. It is well described that obesity exhibits low-grade inflammatory characteristics and that adipose tissue is subject to a continuous remodeling process, establishing dynamic effects. A possible hypothesis for this difference pointed out by the reviewer would be due to arginase (Arg1) which in obesity is upregulated and competes with Nos2 for L-arginase, resulting in a decrease in NO, impairing adipocyte function and increasing macrophage infiltration in the tissue. N7 seems to exert an important action preventing the Nos2 gene expression, corroborating the same already published effect in both inguinal fat depot (Front Endocrinol. 2020 Sep 30;11:537061. doi: 10.3389/fendo.2020.537061) and bone marrow macrophages (J Biol Chem. 2015 Jul 3;290(27):16979-88doi: 10.1074/jbc.M115.646992) of mice fed an high fat diet.
- Mcp1, Tnf-a, IL-6, Why no change in HFD plus n7 and HFD of Epi-WAT but these three decreased in Epi-SVF????
Response: In the present work, it was not possible to test the effect on n7 in isolated adipocytes, only on Epi-SVF and in whole Epi-WAT, and also in 3T3-L1 cells. We have previously demonstrated the huge contribution of mature hypertrophied adipocytes in the production of Mcp1, Tnf-a, IL-6 (Nutrients. 2021 doi: 10.3390/nu13030754; Physiol Rep. 2020 doi: 10.14814/phy2.14380; J Physiol. 2016 doi: 10.1113/JP272541). We believe that this major contribution should outweigh and mask the important effect of n7 on cells from FEV.
It is well described that the FEV cells can modulate the environment and have potential as a regenerative therapy for several complications, effectively impacting the maintenance of homeostasis. Herein, we demonstrated that n7 was able to inhibit the production of these pro-inflammatory cytokines (Mcp1, Tnf-a, IL-6), illustrating how promising the study of n7 action on FEV cells can be.
- Figure 4: Mcp1, Tnf-a, IL-6, Cxcl10, Nos2 can be seen n7 effects. Why IL-6 in SVF was very low mRNA expression???
Response: It is important to emphasize that statistically, the Il-6 data were not lower than the controls. Our results corroborate other studies showing that palmitoleic acid decreases the expression of Il-6 in vascular endothelium (Mol Nutr Food Res. 2018 Oct;62(20):e1800322.doi: 10.1002/mnfr.201800322), and exerts a potent inhibitor effect in its expression in wound tissues, acting as an anti-inflammatory agent (PLOS ONE 2018. October 11,doi.org/10.1371/journal.pone.0205338 October 11, 2018).
- Figura 5: From 3T3-L1 data showed that all tested mRNAs are no significant difference between n7 and PA, what that mean??? How about the protein levels of cytokines???
If 3T3-L1 cells pretreated with PA for certain time and next addition of n7 as one more group PA plus n7, this protocol is similar to the mice treatment,
We thank for all criticism and suggestion that are very appropriate.
Anyway, since n7 is also not different from control, it means that only the saturated fatty acid PA (16:0), but not unsuturated (16:1n7), increases de proinflammatory cytokines expression. Different results were found from 3T3L1 to adipogenic markers, where N7 alone was able to promote an increase in mRNA expression of the master regulators of adipogenesis (that codes early and late adipogenic markers) – Figure 2.
The association PA plus n7 was not performed in this work, such asthe protein levels of cytokines. We agree with the Reviewer about the importance,which makes us think of a future project to conduct this experiments. Anyway, we are confident that the results obtained and showed in the first version support the main hypothesis of this study.
- Line 234: changed to “corroborating the results, we also detected an increment in the expression of ”...
- Line 261: IL-6, IL-10, Tnf-a and Mcp-1 increased???
Response: The reviewer is right.
Looking at Epi- (whole) WAT, our data showed that HFD induced an increase of Mcp1, Tnfa, IL-6 and Nos2 mRNA content, but we did not observe any considerable differences when comparing the HFD and HFD+n7 group, except for a reversal of Nos 2 and an increment in Il10 mRNA levels in animals treated with n7.
- Line 326: sacrificed instead of killed – changed
- Line 347:80oC – changed to 80o C
- Line 362: 100μM – changed to 100 μM

Round 2
Reviewer 1 Report
The authors answered most of the raised questions. My recommendation is "Accept in the present form".
Reviewer 2 Report
This new version ms is promptly improved. I do agree that this revised ms can be accepted for next step.